# The Relevance of Pharmacokinetic Biomarkers in Response to Methadone Treatment: A Systematic Review

**DOI:** 10.3390/ph18050623

**Published:** 2025-04-25

**Authors:** Sheila Recarey-Rama, Jesús Gómez-Trigo, Almudena Gil-Rodriguez, Eduardo Dominguez, Inés Sánchez-Martínez, Ángela Riveiro-Recimil, Alba Barral-Raña, Jose de Leon, Ana Rodriguez-Viyuela, Manuel Arrojo, Angel Carracedo, Olalla Maroñas

**Affiliations:** 1Genomic Medicine Group, Center for Research in Molecular Medicine and Chronic Diseases (CiMUS), University of Santiago de Compostela, 15782 Santiago de Compostela, Spain; sheila.recarey.rama@usc.es (S.R.-R.); jesusgomeztrigo@gmail.com (J.G.-T.); almudena.gil@usc.es (A.G.-R.); eduardo.dominguez@usc.es (E.D.); alba.maria.barral.rana@sergas.es (A.B.-R.); anarovi15@gmail.com (A.R.-V.); angel.carracedo@usc.es (A.C.); 2Pharmacogenomics and Drug Discovery Group (GenDeM), Health Research Institute of Santiago de Compostela (IDIS), 15706 Santiago de Compostela, Spain; 3Psychiatry Service, University Hospital Complex of Santiago de Compostela, Galician Health System (SERGAS), 15706 Santiago de Compostela, Spain; manuel.arrojo.romero@sergas.es; 4Faculty of Pharmacy, University of Santiago de Compostela, 15705 Santiago de Compostela, Spain; ines.sanchez.martinez@rai.usc.es; 5Genetics Group, Health Research Institute of Santiago de Compostela (IDIS), 15706 Santiago de Compostela, Spain; angelariveiro.recimil@usc.es; 6Mental Health Research Center at Eastern State Hospital, University of Kentucky, Lexington, KY 40511, USA; jdeleon@uky.edu; 7Neuroscience Investigation Group (CTS-549), Neuroscience Institute, University of Granada, 18016 Granada, Spain; 8Centre for Biomedical Network Research on Mental Health (CIBERSAM), Santiago Apóstol Hospital, University of Basque Country, 01004 Vitoria, Spain; 9Psychiatric Genetics Group, Health Research Institute of Santiago de Compostela (IDIS), 15706 Santiago de Compostela, Spain; 10Galician Public Foundation of Genomic Medicine (FPGMX), Galician Healthcare Service (SERGAS), 15706 Santiago de Compostela, Spain; 11Center for Biomedical Network Research on Rare Diseases (CIBERER), Instituto de Salud Carlos III, 28029 Madrid, Spain

**Keywords:** metabolism, genes, methadone, biomarkers, opioid addiction, maintenance therapies, pharmacogenetics

## Abstract

**Background/Objectives**: Methadone maintenance treatment (MMT) is widely used in opioid use disorder (OUD). Its efficacy is influenced by its metabolism, primarily mediated by Cytochrome P450 (CYP450) enzymes in the liver. Genetic polymorphisms in CYP450 genes and other factors, such as age, sex, and concomitant treatments, contribute to interindividual variability in methadone response. This article addresses the relevance of pharmacokinetic biomarkers in methadone metabolism and its impact on treatment outcomes in European populations over the past 25 years. **Methods**: A systematic review was conducted using four databases (PsycINFO, PubMed, Scopus, and Web of Science) for studies published between 2000 and 2024 following the PRISMA 2020 guidelines (CRD42025641373 in PROSPERO). Two independent reviewers screened and assessed the study quality using NHLBI tools. Discrepancies were solved through consensus. Relevant data including sample size, genetic biomarkers, and key findings were extracted for each study. Data were synthesized and described in detail. **Results**: Fourteen studies on pharmacogenetic biomarkers influencing methadone metabolism in European populations were analyzed, encompassing a total of 3180 subjects. *CYP2B6**6 was identified as a key variant associated with increased (S)-methadone plasma levels, potentially leading to cardiac complications, while the role of other pharmacokinetic genes, including *ABCB1* and *CYP2D6*, was inconclusive. **Conclusions**: Genetic polymorphisms significantly influence methadone metabolism, with the *CYP2B6**6 allele playing a key role in (S)-methadone metabolism and associated with cardiac risks. Pharmacogenetic studies integrating co-mediation—the principal cause of phenoconversion—as a potential variable alongside gender differences and encompassing adequate sample sizes could improve outcomes and establish the basis for personalized medicine of MMT.

## 1. Introduction

Addiction is a chronic disease with a neurological basis characterized by relapses [1,2]. In this line, the neurobiological foundations of craving, a key component of dependence syndromes and considered as a central element of the motivational drive in addiction, is considered an interesting research field [3]. Furthermore, craving is a common symptom across various substance use disorders, including those related to alcohol, nicotine, cannabis, cocaine, and other psychoactive drugs [4]. In this context, several theories have been proposed to explain different aspects of neuroadaptation in addiction, including the opponent process theory, inhibitory control theory, reward deficiency theory, incentive sensitization theory, aberrant learning theory, and anti-reward theory [5].

Opioid use disorder (OUD), also known as opioid addiction, is a pervasive disease characterized by the compulsive or uncontrollable seeking and consumption of opioids. OUD represents a health-related problem with social and economic consequences for the individual and the entire society, deteriorating convivence and quality of life [6]. Metabolic Syndrome (MetS) is a complex condition that can arise in patients with OUD as a result of increased calorie consumption, decreased physical activity, or a combination of both factors [7]. Over the past decade, the prevalence of this disorder has increased. It was reported in 2017 that 0.6% of the Spanish population between 15 and 64 years old had consumed heroin at some time, and the number of people using opioids worldwide reached 60 million in 2022 [8,9].

This continuous growth in the number of opioid consumers highlights the necessity of understanding the mechanisms underlying this disorder. Related to this, it is known that the transition from a sporadic user of opioids to an addict is surrounded by risk factors from different sources, which include social, developmental, behavioral, and genetic elements [6,10,11,12,13]. Moreover, people with OUD usually suffer from other psychiatric and medical comorbidities, the most common being anxiety and depression [8,14]. These comorbidities could complicate its diagnosis and exacerbate related symptoms, such as cravings and withdrawal symptoms, which may have an impact on its treatment and prevention [15].

Therapeutic approaches for OUD management encompass both behavioral and pharmacological, the most effective strategy being the combination of both. Whereas behavioral techniques, such as Cognitive Behavioral Therapy (CBT) and Contingency Management, are based on reinforcing patients’ motivation and improving attitudes to confront substance use, pharmacological treatments consist in the prescription of drugs (such as methadone, buprenorphine, and naltrexone) to overcome related symptoms and prolong detoxification once achieved [4,6,16]. The most studied drug for OUD treatment is methadone [17].

Methadone is a synthetic opioid that acts as a full agonist of the µ-opioid receptor (MOR) and as an agonist of the N-methyl-D-aspartate (NMDA) receptor [18,19]. Methadone has potential serotonergic effects with serotonin and noradrenaline reuptake inhibition and high affinity for serotonin receptors (5-HT2A and 5-HT2C) [20]. It was developed in Germany during World War II to be used as an analgesic drug when it was not possible to obtain morphine. Later, methadone was used as a maintenance treatment for heroin dependence in the mid-1960s and then repurposed to treat OUD with great success [21]. Methadone is commercially available in different formulations, from oral to intravenous, as a racemic mixture composed of two enantiomers: (R)-methadone and (S)-methadone. This chimerism results in different properties, with (R)-methadone presenting a 10-fold higher affinity to MOR in vitro and, as a consequence, higher therapeutic efficacy in vivo [22,23]. Methadone maintenance treatment (MMT) can be administered orally or intravenously, and doses can vary depending on the patient’s level of addiction.

Methadone is biotransformed primarily in the liver by the monooxygenases of the Cytochrome P450 (CYP450) family, which play a key role in the first-pass metabolism. Specifically, it is first metabolized into an unstable metabolite, after N-demethylation and subsequent cyclization, called 2-ethylidene-1,5-dimethyl-3,3-diphenylpyrrolidine (EDDP), which is quickly transformed into 2-ethyl-5-methyl-3,3-diphenyl-1-pyrroline (EMDP) due to its high instability [24]. In this line, it is worth mentioning that some CYP450 enzymes present stereo-selectivity for a specific enantiomer. It has been reported that CYP2C19, CYP3A7, and CYP2C8 enzymes have a preference for (R)-methadone, whereas CYP2B6, CYP2D6, and CYP2C18 metabolize (S)-methadone in the first place. Other enzymes, such as CYP3A4, are not stereo-specific [23,25,26] (Figure 1).

The relationship between the methadone dose, blood concentration, and clinical effect is complicated to assess due to pharmacological interactions and individual differences in terms of pharmacokinetics. For example, elevated plasma levels of (S)-methadone have been associated with cardiac complications, especially QT cardiac (QTc) interval prolongation, which may lead to arrhythmias in patients undergoing methadone maintenance treatment (MMT). In this line, genetic polymorphisms, such as single nucleotide polymorphisms (SNPs), present in genes involved in biotransformation and clearance of methadone, can cause different responses between patients [27,28,29,30]. It has been previously reported by Richards-Waugh et al. in 2014 that the following SNPs of the gene *CYP3A4*, rs2242480 and rs2740574, were overrepresented in patients suffering from methadone overdose fatalities compared with controls [31]; on the other hand, Yang et al. published a Genome-Wide Association Study (GWAS) in 2016, which highlights the association between *CYP2B6*, *SPON1*, and *GSG1L* haplotypes and (S)-methadone plasma concentration [32].

Considering the previous results, genetic polymorphisms bring complexity into the methadone prescription process, together with other factors such as age, body weight, liver and kidney function, as well as concomitant treatments. Methadone has been associated with serotonin toxicity when taken in concomitance with other serotonergic medicines; however, the risk appears low [33]. However, special attention should be given to serotonergic drugs, such as pethidine, monoamine oxidase inhibitors (MAOIs), selective serotonin reuptake inhibitors (SSRIs), serotonin–norepinephrine reuptake inhibitors (SNRIs), and tricyclic antidepressants (TCAs) administered concomitantly in order to avoid the risk of serotonin syndrome [34]. Recognizing these variants and elucidating their impact on methadone maintenance treatment (MMT) could help prevent a lack of effectiveness and the appearance of treatment-related toxicities, decreasing methadone dropout rates [35,36]. Methadone maintenance treatment (MMT) can be administered orally or intravenously, and doses can vary depending on the patient’s level of addiction. The Spanish Agency for Medicines and Health Products (AEMPS) recommends starting with a dose of 10 to 30 mg/day, and based on clinical progress, this dose may be increased to 40 to 60 mg/day over a period of one to two weeks. The final maintenance dose typically ranges from 60 to 100 mg/day, with increments of 10 mg/day each week [37].

To identify pharmacogenetic biomarkers involved in methadone pharmacokinetics and clarify their relationship with the outcome of the treatment in different patients, a systematic review was performed to compile the evidence published over the last twenty-five years (2000 to 2024, both included) on European population studies.

## 2. Materials and Methods

### 2.1. Desing and Registration

This systematic review has been designed following the Preferred Reporting Items for Systematic Reviews and Meta-Analyses (PRISMA) 2020 Guidelines [38]. A specific protocol has been developed for this purpose, which is available in the PROSPERO repository of systematic reviews, with the goal of ensuring transparency and methodological rigor. This systematic review has been registered under number CRD42025641373 since February 2025.

### 2.2. Search Strategy

A systematic literature search was performed across the scientific electronic databases PsycINFO, PubMed, Scopus, and Web of Science. The sequence of search included the following terms, separated using the Boolean operators “AND” and “OR”: (methadone) AND (metabolism) AND (gene* OR polymorphism* OR biomarker*). Syntex adjustments were included when necessary for each database (see Appendix A for full search query syntax). As the automatic filters for the year and language of publication were available in all the beforementioned platforms, the search strategy involved English articles published during the last twenty-five years (2000 to 2024, both included).

### 2.3. Inclusion Criteria

This systematic review was restricted to research studies involving human subjects developed in European countries (Caucasians), which analyzed the impact of polymorphisms related to methadone’s metabolism (pharmacokinetics). Other types of articles, such as reviews, systematic reviews, letters, or similar, were excluded. Adult patients were selected regardless of sex, and the main pharmacological treatment to overcome opioid addiction in each patient should be methadone. Additionally, data from patients presenting special circumstances, such as pregnancy, were not extracted.

### 2.4. Screening Process and Study Selection

Two independent reviewers, S.R.-R. and J.G.-T., were responsible for compiling all the results from each database. J.G.-T. is a psychiatrist with expertise in addictions, while S.R.-R. is an expert in pharmacogenetic biomarkers in psychiatry, particularly in CYP450 genes. We considered the combination of their expertise to be a strong synergy for generating high-quality data for this systematic review. After duplicate elimination, titles and abstracts were revised in accordance with the defined eligibility criteria for first inclusion. Once this phase was completed, a full-text screening review was assessed in order to confirm the ethnicity of participants and the polymorphisms included in each article, as these aspects were not always specified in the title/abstract. When inconsistencies or disagreements regarding the articles included appeared, a meeting was held with the rest of the review team to reach a consensus.

### 2.5. Assessment of Study Quality

All studies meeting the inclusion criteria for full extraction were evaluated through a methodological quality assessment. The risk of bias was assessed independently by two reviewers (S.R.-R. and E.D.) using the Study Quality Assessment Tools designed by the National Heart, Lung, and Blood Institute (NHLBI) [39]. The questions assessed were different depending on the type of study: in the case of cross-sectional studies, the NHLBI Study Quality Assessment Tool for Observational Cohort and Cross-Sectional Studies was used; in the case of case-control studies, the NHLBI Study Quality Assessment of Case-Control Studies was selected. Disagreements or inconsistencies regarding the rated quality were discussed with the rest of the review team.

### 2.6. Data Extraction

S.R.-R. and J.G.-T. performed independent data extraction and compiled this information into an Excel spreadsheet template, which was shared with the rest of the review team. The following information was extracted from each included article: (i) sample size, (ii) mean age, (iii) percentage of Caucasians or recruitment country, (iv) pharmacokinetic-related genes analyzed, (v) main findings, and (vi) association *p*-values. Appendix A were revised to complete this information when necessary.

### 2.7. Data Analysis

The findings of this systematic review were synthesized in narrative (descriptive) form. Statistical association values were extracted from each included study, and the impact on methadone metabolism and clinical outcomes were descriptively assessed. Due to the presence of high heterogeneity of the articles finally included, it was not possible to perform a quantitative analysis. Additionally, no meta-analysis and no subgroup or subsect analyses were undertaken.

## 3. Results

Figure 2 summarizes the selection process of the articles included in this systematic review, according to the PRISMA statement. From the 1044 initial articles obtained from the four scientific databases (PsycINFO, PubMed, Scopus, and Web of Science), 206 records were removed due to duplication. During title and abstract screening, a total of 783 articles did not meet inclusion criteria for first-phase inclusion, leading to 55 articles for full-text assessment. Finally, 14 articles were included in this systematic review.

The designed search strategy identified a total of 1044 articles from the four scientific databases chosen (PsycINFO, PubMed, Scopus, and Web of Science). After the removal of duplicates (206 records), title/abstract screening was performed, leading to the exclusion of a total of 783 articles for not meeting inclusion criteria. Full-text assessment ended up with 14 articles for final inclusion in this systematic review.

### 3.1. Characteristics of Included Articles

The screening process led to fourteen articles finally being included in this systematic review (Table 1). These articles, published between April 2001 and January 2021, could be categorized as cross-sectional studies (*n* = 10.71%) and case-control studies (*n* = 4.29%). All studies were carried out in European countries, except two that took place in the United States. Despite this difference, these articles were finally included, as most participants were Caucasians. Heterogeneity was present across the included studies as the evaluation of the impact of genetic polymorphisms was performed using different exposures: methadone plasma level concentrations, durability of QTc intervals, satisfaction with MMT, risk of death for opioid use, and treatment side effects.

### 3.2. Patients Included

The fourteen articles included in this systematic review analyzed a total of 3180 subjects, of which 709 were controls. The mean age was 3562 years, and the percentage of male subjects was higher than females in all articles except in Lötsch and collaborators (25 males and 26 females) [42]; in the case of Dobrinas et al., data segregated by sex were not available [48]. Additionally, Dobrinas et al.’s study used a different formulation for methadone: instead of administering the racemic mixture as in the rest of the studies, participants received levomethadone, which only contains the (R)-methadone enantiomer. Recruited participants from the 14 studies encompass a total of seven countries (Switzerland (*n* = 5), Germany (*n* = 2), Spain (*n* = 2), United Kingdom (*n* = 1), Denmark (*n* = 1), France (*n* = 1), and the United States of America, USA (*n* = 2)). Selection criteria include Caucasian/European patients, and in the case of the articles from the USA, the lowest percentage of Caucasians was detected in the study performed by Carlquist et al. (74%) [50].

### 3.3. Assessment of Risk of Bias

Specific quality assessment scales were used to evaluate the risk of bias in this systematic review according to the study type (see Appendix A). In reference to cross-sectional studies (*n* = 10), a total of 14 questions were considered for assessing the risk of bias. Regarding sample size justification (question 5), most publications (9 out of 10) lacked justifications and equal inclusion criteria. It should also be noted that question 10, related to the number of times of exposure assessment, is marked negatively in this type of article. Concerning case-control studies (*n* = 4), sample size justification was also not included in most of the articles (3 out of 4), and an absence of similarity in some case-control populations was detected. Additionally, it was not always possible to find out whether the assessors were blinded to the exposure status of participants, with only one study reporting this aspect. Finally, unlike in cross-sectional studies, in which 9 out of 10 confounding variables were measured, most case-control studies (3 out of 4) did not specify if statistical corrections based on confounding variables (such as gender or concomitant medication) were performed.

### 3.4. Genetic Variants Analyzed

This systematic review obtained information for a total of ten relevant genes involved in methadone metabolism. Some of them showed interesting association results for genetic variants in relation to different exposures to methadone.

#### 3.4.1. *CYP2B6* Gene

Located on chromosome 19 (19q13.2), this gene encodes the CYP2B6 enzyme [54]. Regarding genetic variability, a total of forty-nine star alleles (*) have been identified for this gene [55], and six were analyzed in the articles reviewed. It is important to mention that *CYP2B6**6/*6 diplotype was found to be associated with higher (R,S)-methadone plasma level concentrations (*p* = 0.006 [41], *p* = 0.013 [44]). When comparing both enantiomers separately, the association is presented in the case of (S)-methadone, reported by three articles (*p* = 0.0004 [41], *p* = 0.0001 [43], and *p* = 0.0004 [44]); however, (R)-methadone seems to not be affected equally (*p* = 0.14 [41], *p* = 0.18 [44], and *p* = 0.07 [43]), underlying possible stereo-selectivity for (S)-methadone. In line with this, the presence of the *CYP2B6**6 allele has been reported to not have a significant association with a high methadone maintenance dose (*p* = 0.89 [49]), which could be explained by the main therapeutic activity relying on (R)-methadone. In terms of the risk of death, *CYP2B6**6 presence was related to higher methadone concentration in post-mortem samples [46].

Other *CYP2B6* alleles, such as *4, *5, *7, *9, *11, and *26, were also assessed and exhibited controversial results. The study of Dobrinas and coworkers highlighted that the *CYP2B6**5 allele was overrepresented in the low (S)-methadone plasma level group (*p* = 0.005), whereas CYP2B6*9 was underrepresented (*p* < 0.05). Regarding *CYP2B6**11, it was overrepresented in the high (S)-methadone plasma level group (*p* = 0.006). On the other hand, Ahmad et al. found that the *CYP2B6**5/*5 genotype was associated with a significant increase in methadone blood concentration (*p* = 0.002), and *CYP2B6**2 presence was related to a decreased metabolic ratio [52]. Other studies reported no association for these alleles [42,47,49,51].

#### 3.4.2. *ABCB1* Gene

This gene, located on chromosome 7, encodes the ABCB1 membrane-associated protein, also called P-glycoprotein, which belongs to the ATP-binding cassette (ABC) transporters [56]. Variants in this gene were widely studied in this systematic review, although with certain controversies. Crettol et al. reported lower levels of (R,S)-methadone, (R)-methadone, and (S)-methadone to be associated with rs1045642 (*p* = 0.01, *p* = 0.03, and *p* = 0.01, respectively) and rs9282564 (*p* = 0.01, *p* = 0.02, and *p* = 0.01, respectively), whereas rs2032582 was only associated with lower (R)-methadone levels (*p* = 0.04) [43]. According to Iwersen-Bergmann et al., rs1045642 homozygous patients presented higher medulla/blood methadone concentration ratios compared to heterozygous and non-carriers (*p* = 0.0019 and *p* = 0.0042, respectively), while rs1128503 and rs2032582 did not present any association (*p* = 0.5 and *p* = 0.12, respectively) [53]. On the other hand, SNP rs9282564 was underrepresented in a cohort of deceased patients due to opioid addiction in comparison with a cohort of living patients with opioid addiction, which has been reported to underly a possible protective role in the risk of death due to its consumption [51]. Other SNPs did not present any association with MMT [42,47,49,51].

#### 3.4.3. *CYP2D6* Gene

The *CYP2D6* gene codes for the CYP2D6 enzyme, which is involved in the metabolism of many psychotropic drugs [57]. Located in position 22q13.2, *CYP2D6* is a highly polymorphic gene, currently with over 170 star alleles (*) identified in PharmVar [55,58]. Eap and collaborators observed different (R)-methadone concentrations between ultrarapid metabolizers (UMs) and poor metabolizers (PMs) (*p* = 0.009); however, the association was not significant when compared with normal metabolizers (NMs) (*p* = 0.055 and *p* = 0.064, respectively) [40]. Crettol et al. described that UMs showed lower concentrations of (R,S)-methadone and (S)-methadone compared to NMs and intermediate metabolizers (IMs) (*p* = 0.03 and *p* = 0.04, respectively) but not with (R)-methadone (*p* = 0.08) [43], whereas the association for (R)-, (S)-, and (R,S)-methadone was reported by Fonseca et al. (*p* = 0.002 (R), *p* < 0.001, and *p* = 0.048) [47]. Additionally, Pérez de los Cobos et al. found that UM males reported lower satisfaction with MMT than females (*p* = 0.022 [45]). Other studies did not find statistically significant associations for other *CYP2D6* variants, such as *2, *3, *4, *5, *10, and *11 alleles [42,51] or copy number variations (CNVs) [49]. The study by Lötsch et al., which investigated the influence of *CYP2D6* polymorphisms alongside other genetic variants, did not find a significant association with levomethadone plasma concentrations [42]. Other studies, such as Eap et al. (2001) and Crettol et al. (2006), also examined the impact of different CYP2D6 metabolizer types on (R)-methadone, although statistical significance was not always achieved [40,43]. In this context, the authors suggested that it would be worthwhile to continue exploring these variants, including other genes such as ABCB1, even though the current results remain inconclusive.

#### 3.4.4. Other Assessed Genes: *CYP3A4, CYP3A5, CYP1A2, CYP2C8, CYP2C9, CYP2C19,* and *UGT2B7*

Regarding the *CYP3A* gene family (*CYP3A4* and *CYP3A5*), Crettol et al. reported that *CYP3A4**1B’s presence was associated with higher (S)-methadone plasma levels but observed no significance with (R) enantiomer or the mixture of both [43]. Any association was reported for *CYP3A5**3 [42,47,49,51].

Regarding the *CYP2C* family of genes (*CYP2C8, CYP2C9,* and *CYP2C19*), Carlquist and collaborators found that *CYP2C19**2 carriers presented increased dose-corrected concentrations of EDDP, (S)-EDDP, and (R)-EDDP (*p* < 0.005, *p* < 0.004, and *p* < 0.003) [50]. However, *CYP2C19**2 did not have an impact in other studies, in line with alleles *3, *4, and *17 [41,42,43,47,49,51]. Described alleles in the *CYP2C9* gene (*2 and *3) and in *CYP2C8* (*3 and *4) were not correlated with MMT outcomes either [41,42,43,47].

Finally, the impact of polymorphisms in *CYP1A2* and *UGT2B7* genes was also assessed by some studies in this systematic review, although potential associations were not found [42,43,51].

## 4. Discussion

Response to methadone treatment is characterized by a highly interindividual variability, in which several factors have been reported to play an important role. In order to elucidate the impact of pharmacogenetic polymorphisms on the outcomes of treatment, this systematic review, oriented to collect evidence of genetic biomarkers related to methadone metabolism in Europeans, was developed.

Specifically, ten different pharmacokinetic genes were analyzed across the fourteen articles included, the *CYP2B6* gene being the most studied gene in relation to MMT. Within all *CYP2B6* polymorphisms, the *CYP2B6**6 allele seems to be crucially involved in methadone metabolism, especially in the case of the (S)-enantiomer, findings that are in line with other scientific publications [59,60]. This stereo-selectivity of methadone is important because high (S)-methadone plasma level concentrations were reported to be related to cardiac complications, with the possibility of prolonging the duration of the QTc intervals, leading to arrhythmia problems in patients under MMT [18,19].

The combination of *CYP2B6* haplotypes can be translated into clinical phenotypes, which represent the metabolizer status of each individual [61]. Although forty-nine star alleles for the *CYP2B6* gene have been identified, only six are included in the Clinical Pharmacogenetics Implementation Consortium (CPIC^®^) guideline for genotype-based recommendations for MMT. Concretely, two normal function alleles (*1 and *2), one increased function allele (*4), one decreased function allele (*6), and one no-function allele (*18). The *3 allele has been also mentioned; however, its functionality remains uncertain. Five categories of clinical phenotypes have been established by the Pharmacogenomics Knowledge Base (PharmGKB) and the CPIC^®^: normal metabolizers (NMs) for individuals carrying *1/*1, *1/*2, or *2/*2; rapid metabolizers (RMs) for *1/*4; ultrarapid metabolizers (UMs) for *4/*4; poor metabolizers (PMs) for *6/*6, *18/*18, or *6/*18; and intermediate metabolizers (IMs) for *1/*6 or *1/*18. The CPIC guideline for MMT establishes standard dosing, titration, and drug monitoring for RMs, NMs, Ims, and PMs, whereas genotype-based recommendations for UMs are currently not available [62].

It is also important to consider that, in addition to the individual genetic landscape, the presence of co-medications may contribute to phenoconversion. This phenomenon resulting from inhibition or induction with other drugs causes discrepancies between the patient’s genotype and the real patient’s phenotype, e.g., patients categorized as NMs converted into PMs phenotypically. The presence of potent inhibitors for the CYP2B6 enzyme, such as ticlopidine during MMT, may potentially reduce its metabolic activity and lead to interindividual variability in drug response. Conversely, the combination of MMT with CYP2B6 inducers or specific circumstances, like pregnancy, interfere by increasing its activity, leading to subtherapeutic methadone doses and nonadherence to treatment [63,64,65,66,67]. Therefore, although there are studies addressing the impact of co-medication during MMT [68,69,70], still further studies will need to include co-medication as a potential variable.

The role of the rest of the polymorphisms addressed remains unclear due to incongruencies between the studies included and other publications [29,71,72], thus underlining that the methadone metabolization process is still unknown and other non-metabolic related variants could have an impact on MMT outcomes. PharmGKB [73] compiles in vitro studies that indicate that approximately 63–74% of the drug is metabolized by the *CYP3A4* gene [23]. However, there is still controversy, since many publications point out that the *CYP2B6* gene is likely responsible for most of the metabolization of methadone [74,75].

The influence of other gene variants on methadone plasma levels has also been observed in other studies. The *POR* gene encodes the P450 oxidoreductase enzyme (POR), which plays a key role in electron transfer from the Nicotinamide Adenine Dinucleotide Phosphate (NADPH) to microsomal CYP450 enzymes, such as CYP3A4 or CYP3A5. Variants in this gene could potentially affect methadone metabolism [76,77,78]. It has been also described that polymorphisms in the Pregnane X receptor (PXR) could play a role in regulating CYP2B6 activity [29]. Other publications studied the role of pharmacodynamic variants, which mainly involve the opioid receptor gene (*OPRM1*) [42,46,79]. Therefore, further investigations taking into consideration these genes together might be of interest to create predictive models for methadone prescription.

Another aspect that should be taken into consideration for predictive models is gender differences in terms of methadone response. As reported by Pérez de los Cobos and collaborators, male UMs showed less satisfaction with MMT than female UMs [45]. In addition, other articles have reported that women required higher doses of methadone [80] and confirmed the existence of differences in relapse rates between males and females [81,82], underlining the possible role of sexual hormones in the outcome of MMT. Although the risk of bias assessment did not show the low methodology quality articles included in this review, the lack of balanced sex representation (only one of the articles presented similar sample sizes between males and females) may introduce the risk of bias. Therefore, findings might not accurately reflect the outcomes for both genders equally and should be taken into consideration.

In addition to the last point, it is important to highlight that this systematic review presents some limitations. On the one hand, some articles used reduced sample sizes, which may compromise the possibility of obtaining significant associations. Additionally, most articles did not include a control group (*n* = 10), which would be helpful to validate the results obtained in each case. Finally, although articles analyzing Caucasian and European patients were selected, the potential risk of bias cannot be excluded since no populational analysis has been performed on individuals; therefore, population stratification cannot be ruled out. Concerning the review process, the heterogeneity of study designs complicates the synthesis of findings. Furthermore, the authors are aware that the absence of subjectivity cannot be guaranteed.

## 5. Conclusions

MMT response varies significantly between individuals, and genetic polymorphisms can explain a part of these metabolic differences. This review highlights the importance of the *CYP2B6**6 allele in (S)-methadone metabolism, encompassing clinical implications due to its association with the risk of cardiac complications. The CPIC^®^ guideline for *CYP2B6* genotype-based recommendations for MMT establishes standard dosing, titration, and drug monitoring for RMs, NMs, Ims, and PMs; however, it is important to take into consideration the co-medication of the patient when translating *CYP2B6* genotypes into the expected phenotypes in order to avoid phenoconversion. Aligning future research with CPIC^®^ nomenclature used in guidelines would not only facilitate consistent interpretation of genetic data across studies but also ensure an alignment with the established framework, facilitating the integration of pharmacogenetics into clinical practice. Concerning the rest of the genes included, inconsistencies have been detected in the role of other pharmacokinetic genes that are potentially relevant. The lack of consensus on the primary CYP450 enzyme responsible for methadone metabolism and the potential influence of pharmacodynamic variants, such as *OPRM1*, suggest that a more systematic approach is needed to optimize MMT. Additionally, gender differences in methadone response indicate that sex-based variations should be considered in future studies and treatment guidelines. Despite methodological limitations, such as small sample sizes and adequacy of control groups, pharmacogenetic research seems to promise an improvement in MMT outcomes. Currently, dosing continues to be primarily adjusted based on the severity of addiction; however, this approach might result in the underdosing or overdosing of the patient, leading to treatment interruptions and increasing both public healthcare system costs and health risks for patients, including relapse into substance use. In this context, this review highlights the potential of implementing genetic testing as a future tool, together with clinical outcomes to optimize dosing, in order to enable a more precise personalization of treatment from the outset.

## Figures and Tables

**Figure 1 pharmaceuticals-18-00623-f001:**
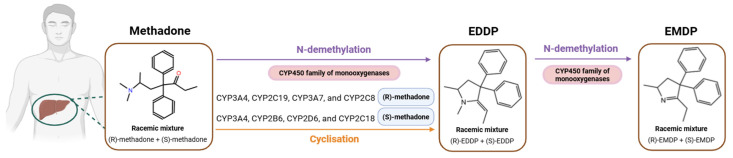
Methadone metabolism in the liver. Created with BioRender (https://app.biorender.com/, accessed on 16 April 2025).

**Figure 2 pharmaceuticals-18-00623-f002:**
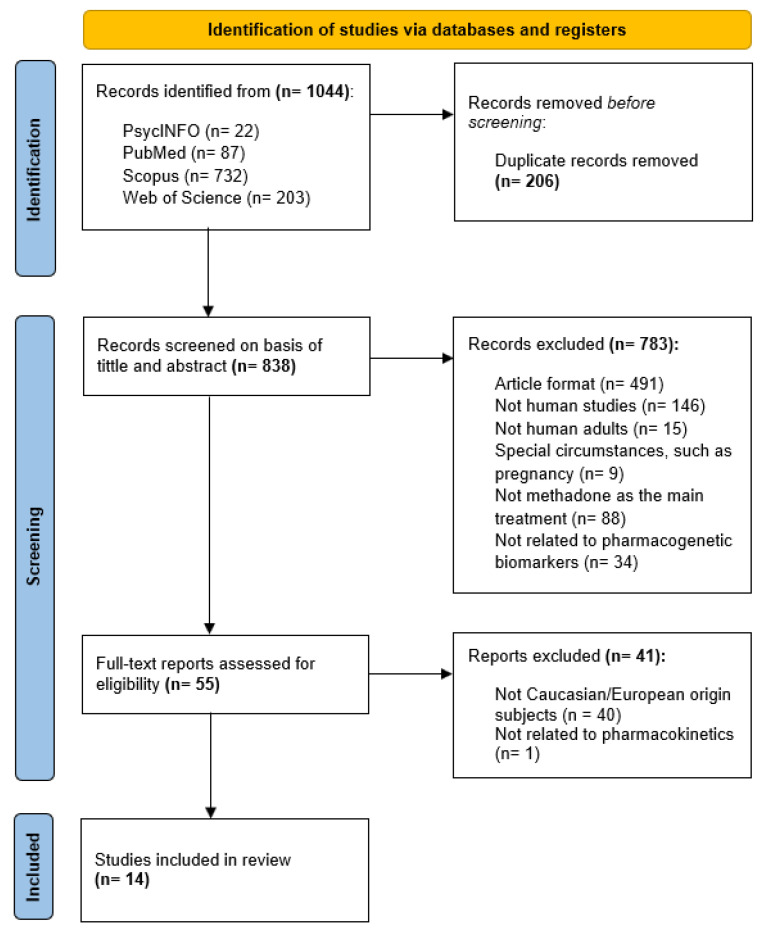
PRISMA flow diagram of study selection.

**Table 1 pharmaceuticals-18-00623-t001:** Summary of findings from the final included studies in this systematic review.

Authors and Year	Sample Size (n)	Mean Age	Ethnicity/Recruitment Country	PK Genes Included	Findings	*p*-Value
Eap et al., 2001 [40]	256	31 (SD not available)	Switzerland	*CYP2D6*	(R)-methadone concentrations to dose–weight ratios differ between UMs and PMs ● (R)-methadone concentrations to dose-to-weight ratios do not differ between UMs and NMs or PMs and NMs	0.009 0.055; 0.064
Crettol et al., 2005 [41]	209	36 ± 8	Switzerland	*CYP2B6*	*CYP2B6* *6/*6 diplotype is associated with higher (R,S)-methadone plasma levels when compared to heterozygous patients and non-carriers *CYP2B6* *6/*6 diplotype is associated with higher (S)-methadone plasma levels when compared to heterozygous patients and non-carriers ● *CYP2B6* *6/*6 diplotype is not associated with higher (R)-methadone plasma levels when compared to heterozygous patients and non-carriers	0.006 0.0004 0.14
				*CYP2C19*	● *CYP2C19**2 and *CYP2C19**3 alleles do not influence methadone plasma levels	>0.05
				*CYP2C9*	● *CYP2C9**2 and *CYP2C9**3 alleles do not influence methadone plasma levels	>0.05
Lötsch et al., 2006 [42]	51	27.2 ± 5.3	Germany	*ABCB1*, *CYP2B6*, *CYP1A2*, *CYP3A5*, *CYP2C8*, *CYP2C9*, *CYP2C19*, *CYP2D6*	The following polymorphisms do not show significant association with side effects of (R)-methadone, measured by change in pupil diameter: • *ABCB1* (rs2032582 and rs1045642) • *CYP1A2* (*1D and *1F alleles) • *CYP2B6* (*4, *5 and *6 alleles) • *CYP2C8* (*3 and *4 alleles) • *CYP2C9* (*2 and *3 alleles) • *CYP2C19* (*2 and *4 alleles) • *CYP2D6* (*3, *4, *5 and *41 alleles) ● *CYP3A5* (*3 allele)	Not available
Crettol et al., 2006 [43]	245	36 ± 8	95% Caucasian, Switzerland	*CYP3A4*	*CYP3A4**1B presence is associated with higher (S)-methadone plasma levels ● *CYP3A4**1B presence is not associated with (R)- or (R,S)-methadone plasma levels	0.048 >0.3, >0.6
				*ABCB1*	rs1045642 carriers are associated with lower levels of (R)-, (S)-, and (R,S)-methadone rs9282564 carriers are associated with lower levels of (R)-, (S)-, and (R,S)-methadone ● rs2032582 carriers are associated with lower levels of (R)-methadone but not with lower levels of (S)- and (R,S)-methadone	0.03, 0.01, 0.01 0.02, 0.01, 0.01 0.04, 0.06, 0.05
				*CYP2B6*	● *CYP2B6**6/*6 diplotype is associated with higher (S)-methadone plasma levels. A similar trend to (R)-methadone can be seen, which is not significant	0.0001, 0.07
				*CYP2D6*	UMs show lower concentrations of (S)-methadone and (R,S)-methadone compared to NMs and IMs ● UMs show a trend of lower concentrations of (R)-methadone compared to NMs and IMs, but it is not significant	0.03, 0.04 0.08
				*UGT2B7*, *CYP1A2*, *CYP2C9*, *CYP2C19*	Other genetic polymorphisms do not exhibit relation with methadone ((S), (R) or (R,S)) plasma levels: • *UGT2B7* (*2A allele) • *CYP1A2* (*1F allele) • *CYP2C9* (*2 and *3 alleles) ● *CYP2C19* (*2 and *4 alleles)	Respectively, >0.09 >0.4 >0.3 >0.1
Eap et al., 2007 [44]	179	36 ± 8	98% Caucasian, Switzerland	*CYP2B6*	*CYP2B6* *6/*6 diplotype is associated with increased (R,S)-methadone plasma concentrations *CYP2B6* *6/*6 diplotype is associated with increased (S)-methadone plasma concentrations *CYP2B6* *6/*6 diplotype is not associated with increased (R)-methadone plasma concentrations ● *CYP2B6* *6/*6 diplotype is associated with increased risk of prolonged QTc interval	0.013 0.0004 0.18 0.017
Pérez de los Cobos et al., 2007 [45]	205	36.8 ± 6	Spain	*CYP2D6*	● Male UMs for *CYP2D6* reported lower satisfaction with methadone treatment than female UMs measured with the Verona Service Satisfaction Scale (VSSS)	0.022
Bunten et al., 2010 [46]	67	31 ± 1.67	United Kingdom	*CYP2B6*	● *CYP2B6**6 presence is associated with high methadone concentration in post-mortem blood samples from individuals, attributed to methadone toxicity	0.039
Fonseca et al., 2011 [47]	105	38 ± 8	Spain	*CYP2D6*	● UMs show higher concentrations of (R)-, (S)-, and (R,S)-methadone compared to NMs	0.002, <0.001, 0.048
				*CYP2B6*	● Although not significant, *CYP2B6**6 allele carriers displayed higher (S)-methadone plasma levels and received lower doses of methadone	>0.05
				*CYP3A5*, *CYP2C9*, *CYP2C19*, *ABCB1*	● Other genotypes do not exhibit significant differences between responders and non-responders to methadone treatment	0.211 to 1.000
Dobrinas et al., 2013 [48]	276	Not available	Switzerland	*CYP2B6*	*CYP2B6**11 allele is overrepresented in the high (S)-methadone level group *CYP2B6**9 allele is underrepresented in the low (S)-methadone level group ● *CYP2B6**5 allele is overrepresented in the low (S)-methadone level group	0.006 <0.05 0.005
Mouly et al., 2014 [49]	81	43.7 ± 8.1	85.2% Caucasian, France	*ABCB1*, *CYP2B6*, *CYP3A5*, *CYP2C19*, *CYP2D6*	The following polymorphisms do not show significant association with methadone maintenance dose: • *ABCB1* (rs1045642) • *CYP3A5* (*3 allele) • *CYP2B6* (*4 and *6 alleles) • *CYP2C19* (*2 and *17 alleles) ● *CYP2D6* (copy number variations)	Respectively, 0.79 0.63 0.93, 0.89 0.2, 0.47 0.19
Carlquist et al., 2015 [50]	42	30.3 ± 9.9	74 % Caucasian, United States of America	*CYP2C19*	● *CYP2C19**2 carriers are associated with increased quantities of dose-corrected concentrations of EDDP, (S)-EDDP, and (R)-EDDP	<0.005, <0.004, <0.003
Christoffersen et al., 2016 [51]	977	42 (SD not available)	Denmark	*ABCB1*	● rs9282564 is underrepresented in the deceased patients with opioid addiction cohort (DOA) compared to the living patients with opioid addiction cohort (LOA)	0.027
				*CYP2B6*, *CYP2D6*, *CYP3A5*, *CYP2C19*, *UGT2B7*	Other genetic polymorphisms are not associated with risk of death due to opioids: • *CYP3A5* (*3 allele) • *UGT2B7* (rs28365063, rs7438135) • *CYP2C19* (*2 and *17 alleles) • *ABCB1* (rs1045642, rs2032582, rs1128503) • *CYP2B6* (*7, *9 and *26 alleles) ● *CYP2D6* (*2, *4 and *10 alleles)	Respectively, 0.596 0.431, 0.714 0.822, 0.082 0.506, 0.741, 0.472 0.318, 0.180, 0.230 0.358, 0.765, 0.609
Ahmad et al., 2017 [52]	380	34.08 ± 11.40	100 % Caucasian, United States of America	*CYP2B6*	*CYP2B6**2 allele is overrepresented in the methadone-only group compared to control individuals ● *CYP2B6**5/*5 patients present a significant increase in blood methadone concentration	0.04 0.002
Iwersen-Bergmann et al., 2020 [53]	107	41 (SD not available)	Germany	*ABCB1*	rs1045642 homozygous patients present higher medulla/blood concentration ratios compared to heterozygous and non-carriers of the variant ● rs1128503 and rs2032582 are not associated with medulla/blood concentration ratios	0.0019, 0.0042 0.5, 0.12

SD: standard deviation; UMs: ultrarapid metabolizers; PMs: poor metabolizers; NMs: normal metabolizers; IMs: intermediate metabolizers; QTc: QT interval of the cardiac cycle; EDDP: ethylidene-1,5-dimethyl-3,3-diphenylpyrrolidene.

## Data Availability

The full protocol of this systematic review is registered and available in PROSPERO under the registration code CRD42025641373. The raw data supporting the conclusions of this article will be made available by the authors on request. Requests to access these datasets should be directed to Olalla Maroñas, olalla.maronas@usc.es.

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
