# Peer review of "The Relevance of Pharmacokinetic Biomarkers in Response to Methadone Treatment: A Systematic Review"

_pharmaceuticals, 2025, doi:10.3390/ph18050623_

Round 1
Reviewer 1 Report
Comments and Suggestions for Authors
Title: The relevance of pharmacokinetic biomarkers in response to methadone treatment: a systematic review
What is the main question addressed by the research?
Authors have performed a systematic literature search review using scientific electronic data- bases PsycINFO, PubMed, Scopus, and Web of Science. Their review is mainly for the relevance of pharmacokinetic biomarkers in methadone metabolism and its impact on treatment outcomes in European populations over the past 25 years. The authors found that CYP2B6*6 allele playing a key role in (S)-methadone metabolism and is associated with cardiac risks. This is because of phenoconversion, and the health care professionals shall note the phenoconversion to avoid toxicities with methadone therapy.
- What parts do you consider original or relevant to the field? What
specific gap in the field does the paper address?
As stated in the beginning, the article is focussing on personalized medicine, therefore it is suited best for pharmacology or Biochemistry part.
- What does it add to the subject area compared with other published
material?
It adds to the safe practice of medicine as phenoconversion is very dangerous and is associated with risks for development of toxicities associated with the drug therapy.
- What specific improvements should the authors consider regarding the
methodology? What further controls should be considered?
The manuscript is drafted well. Methodology is acceptable.
ONLY ON Page 3, line 340 and throughout the text, the “CIPIC ® guidelines” shall be corrected as “CPIC guidelines”.
- Are the conclusions consistent with the evidence and arguments presented?
Were all the main questions posed addressed? By which specific experiments?
Yes the conclusions are acceptable.
- Are the references appropriate? YES
. Any additional comments on the tables and figures and the quality of the
data.
NIL

Author Response
Thank you very much for your valuable comments. As suggested in response to the fourth question, we have updated the text from 'CIPIC® guidelines' to 'CPIC guidelines' in line 382, instead of line 340, due to further revisions made during the review process
Reviewer 2 Report
Comments and Suggestions for Authors
Dear Authors,
The article “The relevance of pharmacokinetic biomarkers in response to
methadone treatment: a systematic review” is a very important review which discovers the influence of genetic polymorphisms on pharmacokinetics of the drug methadone.
The article has high scientific significance in the field of pharmacogenetics and personal medicine. The provided analysis have confirmed an important conclusion that genetic polymorphisms significantly influence methadone metabolism, with the CYP2B6*6 allele playing a key role in (S)-methadone metabolism and associated with cardiac risks.
There are some comments.
- ABCB1 protein is usually called P-glycoprotein. Please, add the second name.
- Are there any SNPs in other ABC-transporters responsible for methadone uptake? Are the any SNPs in pregnane X receptor, CAR? Please, discuss this issue.
- There are many reviews relevant to this article for the last years. These reviews have not been cited and discussed. For example:
- Packiasabapathy S, Aruldhas BW, Horn N, Overholser BR, Quinney SK, Renschler JS, Sadhasivam S. Pharmacogenomics of methadone: a narrative review of the literature. Pharmacogenomics. 2020 Aug;21(12):871-887. doi: 10.2217/pgs-2020-0040. Epub 2020 Jul 24. PMID: 32705966; PMCID: PMC7444627.
- Pharmacogenetics of Methadone for Pain Management in Palliative Care
- Methadone pharmacogenetics in vitro and in vivo: Metabolism by CYP2B6 polymorphic variants and genetic variability in paediatric disposition
- There is a lot of articles about the influence of POR variants on methadone pharmacokinetics. But in this review the information about POR variants is absent.
- It is desirable to give a practical recommendation for the gender-specific pharmacogenetic test which should be done before methadone treatment.
Author Response
1.- ABCB1 protein is usually called P-glycoprotein. Please, add the second name.
Thank you very much for your comment. The following information was added in line 309: “also called P-glycoprotein”.
2.- Are there any SNPs in other ABC-transporters responsible for methadone uptake? Are the any SNPs in pregnane X receptor, CAR? Please, discuss this issue.
Thank you very much for the valuable comment. According the first question, ABCB1 is the most studied gene related to methadone treatment. Other ABC transporters, such as ABCC1 (Multidrug resistance-associated protein 1) or ABCG2 (Breast cancer resistance protein) have been identified to be implicated in opioid transport. However, the direct implication of genetic polymorphisms on the genes that encode for them remains unclear.
Concerning nuclear receptors such as the Pregnane X Receptor (PXR) or the Constitutive Androstane Receptor (CAR), it is worth mentioning that some publications have described its role in CYP450 enzymes induction because of their activation through methadone. Whereas more research is needed to elucidate the impact of the CAR gene polymorphisms on methadone metabolism, Tsai and collaborators published an article presenting PXR gene polymorphisms interaction with other CYP2B6 variants and influence methadone metabolism. We agree that this information should be included in the main text (lines 413-417): “There has been also described that polymorphisms in the Pregnane X receptor (PXR) could play a role in regulating CYP2B6 activity [29]. Other publications studied the role of pharmacodynamic variants, which mainly involve the opioid receptor gene (OPRM1) [42,46,79]. Therefore, further investigations taking into consideration these genes together might be of interest to create predictive models for methadone prescription.”
3.- There are many reviews relevant to this article for the last years. These reviews have not been cited and discussed. For example:
Thank you for the valuable comment concerning the bibliography.
- Packiasabapathy S, Aruldhas BW, Horn N, Overholser BR, Quinney SK, Renschler JS, Sadhasivam S. Pharmacogenomics of methadone: a narrative review of the literature. Pharmacogenomics. 2020 Aug;21(12):871-887. doi: 10.2217/pgs-2020-0040. Epub 2020 Jul 24. PMID: 32705966; PMCID: PMC7444627: Unfortunately, it is not possible to add this reference into the main text because the article is not available in Open Access, thus it is not possible to access the information.
- Pharmacogenetics of Methadone for Pain Management in Palliative Care: in this article it was not an established association between ABCB1 variants and methadone doses, as other studies obtained from the systematic review. This reference was added as part of the discussion in line 401.
- Methadone pharmacogenetics in vitro and in vivo: Metabolism by CYP2B6 polymorphic variants and genetic variability in paediatric disposition: the following information about this article was added in lines 408-412, in response to the next question.
4.- There are a lot of articles about the influence of POR variants on methadone pharmacokinetics. But in this review the information about POR variants is absent.
Thank you for your comment. The P450 oxidoreductase (POR) acts as an electron donor for some cytochrome P450 enzymes. The information about this gene was not obtained from the systematic review because the search strategy was focused on the metabolism of methadone. However, we totally agree that it would be a valuable addition, therefore, information about POR variants was added in the discussion (lines 409-412): “The influence of other gene variants in methadone plasma levels has also been ob-served in other studies. The P450 oxidoreductase (POR) gene (POR) encodes the POR en-zyme which plays a key role in electron transfer from the Nicotinamide Adenine Dinucle-otide Phosphate (NADPH) to microsomal CYP450 enzymes, such as CYP3A4 or CYP3A5. Variants in this gene could potentially affect methadone metabolism [76–78].”
5.- It is desirable to give a practical recommendation for the gender-specific pharmacogenetic test which should be done before methadone treatment.
Thank you for your comment. From our point of view, we are not yet in a position to give a recommendation for a gender-specific pharmacogenetic test. However, in lines 420-423, the text addresses gender differences in response to methadone treatment, possibly influenced by hormonal factors: “In addition, other articles reported that women required higher doses of methadone [80], and confirmed the existence of differences in relapse rates between males and females [81,82], underlying a possible role of sexual hormones in the outcome of MMT.”
Additionally, it is important to have into account that the lack of balanced sex representation in the studies may introduce bias into the results, therefore a reference has been made to the need of considering this variable in future research and clinical guidelines. In lines 423-428, it is added: “Although the risk of bias assessment did not show low methodology quality articles included in this review, the lack of balanced sex representation (only one of the articles presented similar sample sizes between males and females), may introduce the risk of bias. Therefore, findings might not accurately reflect the outcomes for both genders equally and should be taken into consideration.”
Reviewer 3 Report
Comments and Suggestions for Authors
- Neurobiological basis of OUD and psychological factors should be discussed.
- Role of opoid receptors in OUD should be briefly discussed.
- Explain the following sentence
The relationship between methadone dose, blood concentration and clinical effect is complicated to assess due to pharmacological interactions and individual differences in terms of pharmacokinetics.
- Mechanisms of SNRIs, or serotonin-norepinephrine reuptake inhibitors should be discussed.
- Diseases associated with metabolic syndrome in regard to methadone overdose concentration in plasma are advised to be included.
- Biomarkers and their significance should be discussed at the end of introduction.
- Provide the access date and ink of PsycINFO, PubMed, Scopus, and Web of Science.
- This systematic review was restricted to research studies involving human subjects developed in European countries (Caucasians) which analyzed the impact of polymorphisms related to methadone's metabolism (pharmacokinetics).
Do the authors state that polymorphisms affect only Caucasians? Or was it mandatory to exclude other populations from this study?
- Two independent reviewers compiled all the results from each database. Mention their qualification for the relevancy for the generation of data because it is very important study.
- When inconsistencies or disagreements regarding the articles included appeared, a consensus meeting was held with the rest of the review team to reach a consensus. Can authors provide a circular notice number for this meeting?
- All studies meeting the inclusion criteria for full extraction were evaluated through a methodological quality assessment. The risk of bias was assessed independently by two reviewers (S.R.R. and E.D.) using the Study Quality Assessment Tools designed by the National Heart, Lung and Blood Institute (NHLBI).
What was methodological quality assessment for current study? Discuss the Study Quality Assessment Tools designed by the National Heart, Lung and Blood Institute (NHLBI) for this study.
- This study was using a different formulation for methadone: instead of administering the racemic mixture as in the rest of studies, participants received levomethadone, which only contains the (R)-methadone enantiomer.
What was the mode of administration of levomethadone?
- A summary of the results can increase the significance of this study.
- Mean plasma concentrations of levomethadone in different CYP2D6 genotype could be evaluated.
- In the conclusion, authors mentioned that importance of CYP2B6*6 allele in (S)-methadone metabolism, which has clinical implications due to its association with the risk of cardiac complications.
However, the authors did not cite it in the introduction, why?
- What is the standard dose of MMT?
- What is the clinical relevance of this study? Conclusion is very confusing.
Some sentences are very complex so pay attention on syntax.
Author Response
1.- Neurobiological basis of OUD and psychological factors should be discussed.
Thank you very much for the valuable comment, a paragraph related to this issue was included (lines 63-71): “Addiction is a chronic disease with a neurological basis characterized by relapses [1,2]. Thus, exploring the neurobiological foundations of craving, a key component of de-pendence syndromes being considered as a central element of the motivational drive in addiction would be of interest [3]. Furthermore, craving is a common symptom across various substance use disorders, including those related to alcohol, nicotine, cannabis, cocaine, and other psychoactive drugs [4]. In this context, several theories have been pro-posed to explain different aspects of neuroadaptation in addiction, including the oppo-nent process theory, inhibitory control theory, reward deficiency theory, incentive sensiti-zation theory, aberrant learning theory, and anti-reward theory [5].”
2.- Role of opioid receptors in OUD should be briefly discussed.
Thank you very much for this comment. Of course, the role of opioids receptors is very important since they are related to methadone efficacy, however, the main aim of the current systematic review was to assess the relationship between genes involved in methadone metabolism - pharmacokinetic genes - therefore pharmacodynamic pathway was not explored. Anyhow, since polymorphisms in pharmacodynamic genes can also affect methadone treatment outcomes, a small mention has been made in the discussion (lines 414-415): “Other publications studied the role of pharmacodynamic variants, which mainly involve the opioid receptor gene (OPRM1)”.
3.- Explain the following sentence
The relationship between methadone dose, blood concentration and clinical effect is complicated to assess due to pharmacological interactions and individual differences in terms of pharmacokinetics.
Thank you for your attention to this matter. There may be no direct correlation between the methadone dose administered to an individual and their blood concentration due to differences in pharmacokinetics. For example, genetic polymorphisms assessed in this systematic review could affect enzyme activity, either causing methadone to be eliminated too quickly or increasing its plasma levels. Moreover, pharmacological interactions can interfere with these processes, impacting the expected clinical effect of the prescribed methadone dose. This impact of concomitant medication was also discussed in lines 388-394: “It is also important to consider that, in addition to the individual genetic landscape, the presence of co-medications may contribute to phenoconversion. This phenomenon resulting from inhibition or induction with other drugs causes discrepancies between the patient’s genotype and the real patient´s phenotype, e.g. patients categorized as NMs converted into PMs phenotypically. The presence of potent inhibitors for the CYP2B6 en-zyme, such as ticlopidine during MMT, may potentially reduce its metabolic activity and lead to interindividual variability in drug response.”
4- Mechanisms of SNRIs, or serotonin-norepinephrine reuptake inhibitors should be discussed.
Thank you for the comment. The following sentence has been included in lines 100-102: “Methadone has potential serotonergic effects with serotonin and noradrenaline reuptake inhibition and high affinity for serotonin receptors (5-HT2A and 5-HT2C).”.
On the other hand, according concomitant treatments, also a paragraph has been added, lines 142-146: “However, special attention should be given to serotonergic drugs, such as pethidine, monoamine oxidase inhibitors (MAOIs), selective serotonin reuptake inhibitors (SSRIs), serotonin-norepinephrine reuptake inhibitors (SNRIs), and tricyclic antidepressants (TCAs) administered concomitantly in order to avoid the risk of serotonin syndrome [34].”
5.- Diseases associated with metabolic syndrome in regard to methadone overdose concentration in plasma are advised to be included.
Thank you for this consideration. We agree that information about metabolic syndrome is valuable to include in the text. The following sentence has been added as part of the introduction in lines 76-78: “Metabolic Syndrome (MetS) is a complex condition that can arise in patients with OUD as a result of increased calorie consumption, de-creased physical activity, or a combination of both factors [13].”
6.- Biomarkers and their significance should be discussed at the end of introduction.
Thank you for your comment. Findings related to genetic variants affecting methadone metabolism were included in lines 117-121: “In this line, it is worth mentioning that some CYP450 enzymes present stereo-selectivity for a specific enantiomer. There has been reported that CYP2C19, CYP3A7, and CYP2C8 enzymes have preference for (R)-methadone, whereas CYP2B6, CYP2D6, and CYP2C18 metabolize (S)-methadone in the first place. Other enzymes, such as CYP3A4, are not stereo-specific [23,25,26] (Figure 1)”.
7.- Provide the access date and ink of PsycINFO, PubMed, Scopus, and Web of Science.
Thank you for this comment. Access to databases was carried out at the end of January, and the searches were re-run prior to the final analysis of the results, which took place a month and a half later, in March. It has been added to Supplementary Table 1, the URL link to each database, as follows:
1PsycINFO: Access is provided through the library of the University of Santiago de Compostela (https://www.proquest.com/psycinfo/index?accountid=17253&parentSessionId=x0feycV5CrvA1o2Jve%2BCRsj7eswO1TpvPGfquf6EuBc%3D)
2 PubMed Central®: https://pubmed.ncbi.nlm.nih.gov/
3 Scopus: https://www.scopus.com/search/form.uri?display=basic#basic
4 Web of Science: https://www.webofscience.com/wos/woscc/basic-search
8.- This systematic review was restricted to research studies involving human subjects developed in European countries (Caucasians) which analyzed the impact of polymorphisms related to methadone’s metabolism (pharmacokinetics).
Do the authors state that polymorphisms affect only Caucasians? Or was it mandatory to exclude other populations from this study?
Thank you for your consideration of this matter. Authors are aware that polymorphisms affect to all populations, however, we have limited the search to Caucasian population because the genetic factors and environmental influences align more closely with our situation in Spain. We are starting a project devoted to personalized methadone doses, and thus, results obtained from this systematic review would be more applicable to our context, ensuring better relevance and accuracy when considering treatment protocols and patient outcomes in our setting.
9.- Two independent reviewers compiled all the results from each database. Mention their qualification for the relevancy for the generation of data because it is very important study.
Thank you for the comment. This information is, in fact, very relevant, so after your comment we decided to included into the text (lines 187-191): “Two independent reviewers, S.R.R. and J.G.T., were responsible for compiling all the results from each database. J.G.T. is a psychiatrist with expertise in addictions, while S.R.R. is an expert in pharmacogenetic biomarkers in psychiatry, particularly in CYP450 genes. We considered the combination of their expertise to be a strong synergy for generating high-quality data for this systematic review.”.
10.- When inconsistencies or disagreements regarding the articles included appeared, a consensus meeting was held with the rest of the review team to reach a consensus. Can authors provide a circular notice number for this meeting?
Thank you for considering this matter. Once every question of the assessment tool was answered, responses were joined and discussed between both reviewers. In some cases, there was not necessary to discuss with other team members; however, when necessary, a meeting face to face with the team was held.
11- All studies meeting the inclusion criteria for full extraction were evaluated through a methodological quality assessment. The risk of bias was assessed independently by two reviewers (S.R.R. and E.D.) using the Study Quality Assessment Tools designed by the National Heart, Lung and Blood Institute (NHLBI).
What was methodological quality assessment for current study? Discuss the Study Quality Assessment Tools designed by the National Heart, Lung and Blood Institute (NHLBI) for this study.
Thank you for your question. The Study Quality Assessment Tools developed by the National Heart, Lung, and Blood Institute (NHLBI) were created to support researchers, students, and clinicians in evaluating the internal validity of studies included in systematic reviews. These tools are tailored to specific study designs, which is why Supplementary Tables (ST) 2 and 3 present different types of studies: while ST2 refers to cross-sectional studies, ST3 includes case-control studies. Each tool features a distinct set of questions, adapted to the methodological characteristics of the respective study design. The authors chose the NHLBI tools due to their clear, structured workflow and comprehensive guidance, which make them easy to apply consistently.
12.- This study was using a different formulation for methadone: instead of administering the racemic mixture as in the rest of studies, participants received levomethadone, which only contains the (R)-methadone enantiomer.
What was the mode of administration of levomethadone?
Thank you for the valuable comment. The phrase was rewritten in order to highlight that in Dobrinas' article, levomethadone is used, which only contains (R)-methadone, in contrast to the other articles that use the racemic mixture (lines 256-257): “Additionally, the Dobrinas et al, study was using a different formulation for methadone”
13.- A summary of the results can increase the significance of this study.
Thank you for your attention to this matter. Results obtained from the systematic review are summarized in Table 1, which compiles information about each included article: sample size, mean age of participants, ethnicity or recruitment country, the pharmacokinetic genes assessed and the main findings with their corresponding p-values. Additionally, a graphical abstract was supplied.
14.- Mean plasma concentrations of levomethadone in different CYP2D6 genotype could be evaluated.
Thank you for this comment. It has been added the following information in lines 337-344: “The study by Lötsch et al., which investigated the influence of CYP2D6 polymorphisms alongside other genetic variants, did not find a significant association with levometha-done plasma concentrations [42]. Other studies included such as Eap et al. (2001) and Crettol et al. (2006), also examined the impact of different CYP2D6 metabolizer types on (R)-methadone, although statistical significance was not always achieved [40,43]. In this context, the authors suggested that it would be worthwhile to continue exploring these variants, including other genes such as ABCB1, even though the current results remain inconclusive.”
15.- In the conclusion, authors mentioned that importance of CYP2B6*6 allele in (S)-methadone metabolism, which has clinical implications due to its association with the risk of cardiac complications.
However, the authors did not cite it in the introduction, why?
Thank you for this comment. It is true that cardiac complications due to high (S)-methadone concentrations were not indicated in the introduction, and we agree that it is a valuable information that need to be added. It was mentioned in lines 127-130: “For example, elevated plasma levels of (S)-methadone have been associated with cardiac complications, especially QTc interval prolongation, which may lead to arrhythmias in patients undergoing methadone maintenance treatment (MMT).”
16.- What is the standard dose of MMT?
Thank you for your attention to this matter. It has been added the following information in lines 151-155: “The Spanish Agency for Medicines and Health Products (AEMPS) recommends starting with a dose of 10 to 30 mg/day, and based on clinical progress, this dose may be increased to 40 to 60 mg/day over a period of one to two weeks. The final maintenance dose typically ranges from 60 to 100 mg/day, with increments of 10 mg/day each week [37].”
17.- What is the clinical relevance of this study? Conclusion is very confusing.
Thank you for your comment. We realized that the clinical relevance pf the article was not clearly assessed. Therefore, a paragraph has been added in lines 459-465: “Currently, dosing continues to be primarily adjusted based on the severity of addiction; however, this approach might result in underdosing or overdosing of the patient leading to treatment interruptions, increasing both public healthcare system costs and health risks for patients and including relapse into substance use. In this context, this review high-lights the potential of implementing genetic testing as a future tool together with clinical outcomes to optimize dosing, in order to enable a more precise personalization of treat-ment from the outset”.
Round 2
Reviewer 2 Report
Comments and Suggestions for Authors
The article has been imoroved and can be published
Reviewer 3 Report
Comments and Suggestions for Authors
The authors made lot of significant valuable changes in the manuscript. The authors have provided a nicely detailed and thorough response to the comments from the previous review and have addressed most of my all concerns. Hence, this revision has significantly improved the manuscript. In my view, this manuscript can be considered for publication in the present form.